# Social norms vs socioeconomic vulnerability: Gender identity and female labor force participation in Ecuador

Jorge Yepez [1]*, Sara Caria[2]

1 Instituto de Altos Estudios Nacionales, Escuela de Economía Pública y Sectores Estratégicos, Quito, Ecuador, 2 Dipartimento di Comunicazione ed Economia, Università di Modena e Reggio Emilia, Reggio Emilia, Italy

☯ These authors contributed equally to this work.
* yepezeta@gmail.com

## Abstract

Despite recent advances in reducing the gender gap globally, female labor force participation is still lower compared to men's. Discriminatory social norms, such as the one that assigns to men the role of breadwinners, are important drivers of this difference. Building on the social prescription that "a man should earn more than his wife", we explore the effect that relative income potential has on female engagement in the labor market in Ecuador. Results show that women with potentially higher relative income tend to increase their labor force participation (LFP), which is counter-intuitive with respect to the male breadwinner hypothesis. However, we find a differentiated effect according to women's education and income level. More educated women do not increase their LFP, but tend to shift towards more precarious positions (informal or part-time); women with secondary education increase both their LFP and precariousness; women with only primary education, associated to lower income, increase LFP but do not increase precariousness, which is already quite high. These findings suggest that socio-economic characteristics of households condition the adherence to social norms.

## Introduction

Despite remarkable progress over the last decades, the gender gap remains significant globally [1,2]. In the labor market, women show higher unemployment rates, are predominantly employed in jobs with poorer working conditions and still face a large wage gap [3–5].

An important dimension of the gender gap in the labor market is the labor force participation (LFP). In recent decades, women LFP has increased, as the result of lower fertility rates, higher schooling and a more active pursuit of professional careers. Also, demand for female workers in specific sectors (i.e., services,

**Data availability statement:** All replication code and materials necessary to reproduce the results are publicly available at the Open Science Framework (OSF) repository: https://doi.org/10.17605/OSF.IO/K8WRD. The data underlying the results are drawn from the ENEMDU microdata (Encuesta Nacional de Empleo, Desempleo y Subempleo), which are publicly available from the Ecuadorian National Institute of Statistics and Census (INEC). Due to INEC's data use policies, the microdata cannot be redistributed by the authors. Researchers may download the data directly from the official INEC portal at: https://www.ecuadorencifras.gob.ec/estadisticas-laborales-enemdu/. Detailed instructions for accessing and organizing the data are provided in the OSF repository README file.

**Funding:** The author(s) received no specific funding for this work.

**Competing interests:** The authors have declared that no competing interests exist.

healthcare) and improved care leave provisions and childcare subsidies contributed to increase female employment [6–8]. Still, globally, the LFP is 47% for women, compared to 73% for men, the gap being observable across all regions, with varying widths [5]. Discriminatory social norms are important drivers of this difference: traditional norms often assign to men the role of breadwinners, which entails priority consideration in the workplace. Similarly, the same norms establish that women's primary role is motherhood and caregiving within the household, constraining their opportunity to participate in the workforce [3].

Within this broader perspective, a recent stream of inquiry focuses on relative income within households as a factor influencing women's decision to enter (or remain in) the labor market. Bertrand *et al.* [9] build on the idea of identity [10] to explain how social norms shape a gender identity that creates aversion to the possibility of a wife having a higher income than her husband: the male breadwinner norm hypothesis. They find that when a wife's potential income is higher than her husband's, the probability that she engages in labor market decreases: graphically, this is illustrated by a severe drop in the distribution of the share of income earned by the wife to the right of the half, where the wife's earning is higher than the husband's (Fig 1). This suggests that either women exit the labor market or commit themselves to earn less than their husbands.

Empirical studies tend to confirm the male breadwinner norm in advanced economies [9,11]. In contrast, in developing countries, a complementary strategy for women to conform to these norms, while remaining employed, may involve staying in or transitioning to informal employment, as noted by other studies in Brazil and Uruguay [12,13]. The duality of informal and formal work is a hallmark of labor markets in developing economies, where institutional enforcement is often weaker than in wealthier nations; usually, women are overrepresented in informal employment sectors, which are less regulated and often lack social protection [14].

While gender wage gaps in the labor market have been widely studied, also in Latin America and in Ecuador specifically [15–18], the characteristics of relative income within couples are still scarcely explored. To fill this gap, this study builds on the hypothesis that precarious employment may be a channel through which gender norms influence female LFP in Ecuador. We extend previous analysis to include several dimensions of precarity, such as part-time work, short-term contracts, and informal employment, which is a persistent feature of labor market in Ecuador, as in most developing countries, and concerns a considerable share of women (and couples).

This work aims at exploring the male breadwinner hypothesis in Ecuador, i.e., to investigate whether a decrease in female LFP is associated with the situation in which wives earn more than their husbands, through different labor market segments. It uses cross-sectional data from the National Employment, Unemployment and Underemployment Survey (ENEMDU) data set from 2007 to 2022 and focuses on married or cohabiting different-gender couples aged 18–59. Results are heterogenous across different labor market segments and educational levels, suggesting that socio-economic characteristics of households contribute to condition the adherence to social norms.

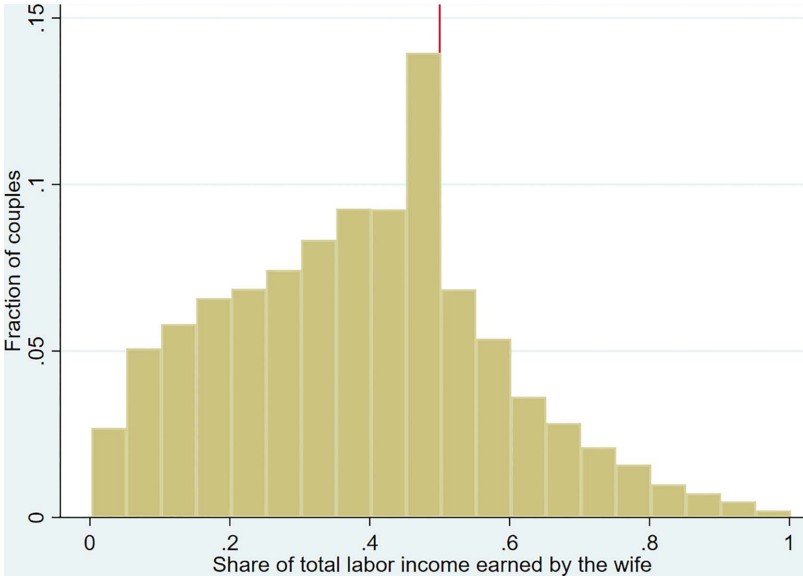

**Fig 1. Histogram of the relative income distribution from 2007 to 2022, with a bin width of 5%.**

The contribution of this work is manifold. First, it is one of the few to explore the association of gender identity with female labor supply in a developing country, with a fragmented labor market structure. Second, it sheds light on how different working conditions, education and income levels may interact with social norms to influence female LFP, consistent with the argument that adherence to social norms is influenced by socio-economic characteristics.

The work is structured as follows. After this introduction, the Literature review section presents an overview of relevant literature. The Data and empirical strategy section describes the dataset and the empirical approach. The Results section presents the findings, and the Conclusions section discusses their implications. Robustness checks can be found in the Appendix in S1 Appendix.

## Literature review

Gender inequalities are still significant in both developed and developing countries; moreover, since the beginning of the 2000s, the pace of convergence seems to be slower than in previous decades [2,19]. Traditional human capital factors (education and experience) alone cannot explain the remaining disparities, especially since women now outperform men in educational attainment and have significantly narrowed the experience gap [20,21]. Consequently, recent research has begun to explore additional elements that may be hindering further progress in labor market equality.

Gender identity roles are often cited to account for patterns that cannot be fully explained by standard economic models. Examples are the persistence of occupational gender segregation [22] and women' underrepresentation in scientific fields [23,24]. Also, it has been argued that gender identities are transmitted within families and society, with relevant repercussions on women's labor market outcomes [25,26] and on the division of household labor [27]. In this perspective, the literature shows a growing interest in inquiring how culture or social norms may influence women' participation in the labor market, i.e., if gender role attitudes can be associated with specific behaviors in the labor supply of women [28,29], exploring drivers and socialization effects [26,30,31], even in particularly restrictive environment for women [32].

Fleche *et al.* [33], in a study in the US, report lower life satisfaction for women who work longer hours than their husbands and suggest that women generally avoid working more than their partners. It is argued that, as primary caregivers

at home, women often value flexible work hours. However, the demand for flexibility, generally associated to lower wages, seems to be influenced by the ability to "afford" such arrangements and may be accessible to women with higher household incomes [34]. Ichino *et al.* [35] provide evidence of how, in Sweden, gender norms shape reallocation of childcare between parents, following changes in their relative earnings: families from more traditional population groups are inclined to reallocate child care between spouses following an increase in husbands' income (due to tax rate reduction in their specific study) but not if the wife's pay increases. Galván and García-Peñalosa [36] investigate the interaction of various identity norms in the US, highlighting that while motherhood seems to have a clear and significant effect on labor supply, the influence of female relative income on LFP, or the so called male breadwinner norm, is not clear.

Bertrand *et al.* [9] focus precisely on the male breadwinner norm, i.e., the prescription that, within a different-gender couple, "a man should earn more than his wife" (p. 572), and explore the influence of gender identity [10] on economic outcomes, particularly LFP. The authors estimate a measure of a woman's potential income, according to the characteristics of specific demographic groups: then, they compare such measure with married women's effective income and find that when a wife's potential income is higher than her husband's, the probability that she engages in the labor market decreases.

Similar methods have been employed to test this gender identity hypothesis in different contexts: results are somehow mixed, but empirical inquiries on this model are still very scarce. Wieber and Holst [11], using data from the German Socio-economic Panel Study, find a significant relation, but only for Western Germany. The authors attribute this peculiarity to the promotion of women as homemakers in the West, when the country was divided. Hederos and Stenberg [37] investigate the male breadwinner norm in Sweden and find a similar pattern, but their study offers a weak support for such hypothesis; the decrease in LFP is small, mainly statistically insignificant and highly concentrated around the one-half of the distribution (when both husband and wife's income is equivalent).

Empirical studies have been concerned mainly with understanding the effect of social norms on economic outcomes in advanced economies. Few attempts have been made to analyze the male breadwinner norm implications in developing countries, where labor markets' structure differs remarkably: findings indicate a more complex relation between gender identity norms and labor market structure. Following Bertrand *et al.* [9], Galvan [13], using the Household Survey on couples in Uruguay from 1986 to 2016, finds results mainly consistent with the male breadwinner norm hypothesis: the higher the probability that the wife earns more than her husband, the less she is likely to participate in the labor market. If employed, she tends to work fewer hours. Also, findings suggest that women respond to gender norms through adjustments in informal employment: some women stay in the workforce but opt for informal jobs with fewer working hours, more flexible and typically paid less. Codazzi *et al.*'s results in Brazil [12] find a negative relationship between female LFP and the probability that a wife earns more than her husband; however, they also find that wives earning more than their husband are more likely to be employed in informal jobs, and interpret this result as a different strategy to conform to gender identity with respect to advanced economies. De Souza *et al.* [38] basically confirms the effect of the male breadwinner norm in Brazil.

Overall, there is a consensus on the significant role that identity norms play in hindering progress towards equality. Women are carrying personal barriers, or "glass ceilings," from their home into the workplace [9]. Overlooking this perspective can lead to an underestimation of the consequences of gender norms on labor market inequalities. A better understanding of the patterns, determinants and implications of female LFP is essential for a richer and more informed policy debate on gender, labor and poverty.

## Data and empirical strategy

Akerlof and Kranton [10] argue that identity and social norms are embedded into broader social contexts, challenging standard economic models, in which preferences are fixed and choices are purely rational. Gender norms refer to what is considered an appropriate behavior for men and women [39], and individuals often adjust their actions to align

 

with these expectations. Often, behaviors, conscious or not, aim at aligning with gender identity, even when it seems counterproductive.

Acting against the norms generates identity-related disutility. One persistent norm is the male breadwinner ideal, i.e., men "should" earn more than their wives, which can influence labor supply and earnings within households [9].

To build intuition, and following Galván [13], Codazzi et al. [12], and Bertrand *et al.* [9], we consider a simple stylized model of female utility. Let a woman's utility be given by:

$$U_i = \begin{cases} a\,w_i - t\,(w_i - h_i) & if\ w_i \geq h_i \\ a\,w_i & if\ w_i < h_i, \end{cases}$$

were $w_i$ and $h_i$ represent the wife's and husband's labor income for couple $i$, respectively; and $a$ and $t$ are parameters. A discontinuity arises when $w_i \geq h_i$, reflecting identity loss from deviating from social norms in which the male partner is expected to earn more. If $t > a$, $U_i$ declines, which may imply that some women may adjust labor supply as their income approaches their husband's. Although highly stylized, this simple model illustrates how social norms can generate behavioral responses around the 50 percent income share threshold, providing the theoretical motivation for our empirical design.

This theoretical prediction motivates examining whether women's earnings distributions exhibit behavioral distortions around the point where wife and husband earn the same.

This study uses cross-sectional data from the ENEMDU (December) dataset spanning 2007–2022, focusing on married or cohabiting different-gender couples aged 18–59 in the coastal and highland regions. The final sample includes 129,031 couples, covering 61% of all households in the dataset.

Ecuador is a high-middle income country with persistent socioeconomic gaps. In 2023, GDP per capita was USD 6.610 [40] and the 2022 poverty rate was roughly 25% [41]. Gender disparities in the labor market are substantial: the pay gap between men and women is about 25% and female labor force participation (LFP) is 52%, compared 86% among men [42]. Precarious work, particularly informality, remains high: in 2023, informal employment accounts for 68% of the workforce, compared to 52% in Latin America and the [5].

The definitions used in this study follow the official ones used by the Instituto de Estadística y Censos (INEC) of Ecuador [43]: *employment in the formal sector* refers to workers employed in establishments registered as taxpayer in the Internal Revenue Service (RUC); *tenured employment* identifies workers holding stable or long-term contracts, as opposed to temporary ones, which may grant social protection, but only for a limited time. Finally, *part-time employment* is defined according to the 36-hour threshold established in Article 4 of Ministerial Agreement 135, published in Official Register 265 on June 19, 2018.

The data reveal substantial gender gaps in labor market outcomes. Male LFP rate is 98%, versus 56% for female. Among those in the labor force, the formal sector employs 51% of men, but only 46% of women and women are significantly more likely to hold part-time jobs (37% vs. 15%).

To explore the possible influence of the breadwinner norm, we created a measure of relative income share:

$$Relative\ Income_i = \frac{w_i}{w_i + h_i}$$

Income is measured using the monthly labor earnings variable provided in the ENEMDU microdata, which includes income from main and secondary occupations and excludes capital income, transfers, and social benefits. We excluded observations where income was 0. Fig 1 shows the relative income distribution. Each bar represents the proportion of couples within a 0.05 relative income bin, with the vertical line marking the 0.5 relative income share. The distribution of female labor income is consistent with the male breadwinner norm, with most couples positioned to the left of 0.5. However, there is a distinct break after the 0.5 threshold. To formally test whether this discontinuity is significant, we conduct

 

McCrary test [44] (Fig A1 detailed in the Appendix in S1 Appendix). In addition, we report year-specific distributions in the Appendix, which display the same qualitative pattern, although statistical significance is reduced due to smaller sample sizes (Figs A2–A5 in S1 Appendix). As a robustness check, we present the same analysis with a 0.01 relative income bin and the corresponding discontinuity test (Figs A6 and A7 in S1 Appendix). The results indicate a statistically significant discontinuous decrease in the wife's share of household labor income after the 0.5 cut-off, suggesting an aversion to situations where the wife earns more than her husband, as proposed by Bertrand *et al.* [9].

Notably, 2.2% of couples are exactly at 0.5, a clustering similar to that documented in Germany [45], and Uruguay [13]. This descriptive evidence may be interpreted as a behavioral adjustment around the breadwinner threshold.

To estimate how gender norms may affect women's labor market outcomes, we adopt the strategy of Bertrand *et al.* [9]. The key explanatory variable is the probability that the wife would earn more than her husband if her income were randomly drawn from the income distribution of observationally similar women. Because earnings are only observed in the realized match, this probability can be used as a counterfactual prediction of whether the breadwinner norm is likely to bind. However, consistent with the discussion in the Conclusions section, this measure may still be affected by sorting and unobserved heterogeneity, so results should be interpreted as descriptive rather than strictly causal.

We construct this probability by defining 48 demographic groups based on age (four age groups: 18–29, 30–39, 40–49, and 50–59 years old); education level (three categories: primary or less, high school, and university); region of residence (Coastal or Highlands); and ethnic group (two categories: mestizo [including white and others] vs. minority groups, including Afro and Indigenous). For each group and year, we compute the 5th to 95th percentiles (p) of women's earnings (in steps of 5 percentiles). For each couple, we compare these potential incomes with the husband's observed earnings and compute the share of percentiles in which the wife would earn more.

This yields an exogenous measure:

$$\textit{Prob. wife earns more}_i = \frac{1}{19} \sum_p 1_{\{w_i^p > h_i\}} \tag{1}$$

We also constructed the same percentiles using only employed women's income. The results are quantitatively the same as those presented in the Results section and can be found in the Appendix in in S1 Appendix.

We then estimate the following linear probability model:

$$y_i = \beta_0 + \beta_1 \textit{Prob.wife earns more}_i + X_i'\beta + \epsilon_i \tag{2}$$

Where $y_i$ is each labor market outcome from Table 1 and $X_i$ includes controls for husband's income, the wife's potential income at each 5th percentile (corresponding to the actual percentile within her demographic group, from the 5th to the 95th), dummy variables for both the wife's and husband's age groups, education levels, and ethnic groups, region and time fixed effects. Observations with missing values for the husband's labor income are excluded from the estimation sample. Standard errors were clustered by the wife's demographic group level to account for potential within-group correlation in labor market outcomes, following Galván [13]. In addition to the baseline regression, alternative specifications are estimated by adding further controls, following the approach of Bertrand *et al.* [9]. Since the husband's income may affect the household's utility in a nonlinear way, we include a cubic polynomial of the husband's income in our model. Additionally, we account for the possibility that women who marry men with lower education and earnings might have characteristics that predispose them to stay out of the labor force in favor of home production or child care. To address this, we include dummy variables for whether there is a child aged 4 years or younger, as well as for whether there is a child aged 5–12 years.

As previously noted, precarity is a defining feature of the Ecuadorian labor market, yet its impact varies significantly based on workers' skill levels. As illustrated in Table 2, high-skilled workers have substantially higher formal employment and tenure, while low-skilled workers face significantly more informality and part-time work.

**Table 1. Employment rates for males and females in married or cohabiting couples (aged 18–59, average 2007–2022).**

|  | Male | Female | Total |
|---|---|---|---|
| (a) By labor force participation |  |  |  |
| Out of labor force | 1.79 | 43.91 | 22.85 |
| Labor force participant | 98.21 | 56.09 | 77.15 |
| Total | 100 | 100 | 100 |
| (b) By contract duration type |  |  |  |
| No tenure | 73.78 | 77.04 | 74.94 |
| Tenure | 26.22 | 22.96 | 25.06 |
| Total | 100 | 100 | 100 |
| (c) By level of formality |  |  |  |
| Not formal sector | 48.7 | 53.85 | 50.54 |
| Formal sector | 51.3 | 46.15 | 49.46 |
| Total | 100 | 100 | 100 |
| (d) By working time |  |  |  |
| Full-time | 85.29 | 63.32 | 77.3 |
| Part-time | 14.71 | 36.68 | 22.7 |
| Total | 100 | 100 | 100 |

Source: Authors' own elaboration based on ENEMDU.

**Table 2. Employment rates for males and females in married or cohabiting couples (aged 18–59, average 2007–2022) by education level.**

|  | Male | Female |
|---|---|---|
| (a) Formal Sector |  |  |
| Primary | 33.31 | 22.14 |
| High school | 65.49 | 54.30 |
| University | 88.48 | 88.18 |
| (b) Tenure or long-term contract |  |  |
| Primary | 14.84 | 9.21 |
| High school | 34.48 | 21.95 |
| University | 50.76 | 52.34 |
| (c) Part-time |  |  |
| Primary | 18.65 | 47.83 |
| High school | 11.07 | 37.72 |
| University | 7.98 | 18.31 |

Source: Authors' own elaboration based on ENEMDU.

This evidence suggests a divergence in how gender norms impact women based on their skill levels. For more educated women, the professional identity tied to their careers may conflict with traditional gender roles, making them less likely to adhere to such norms. In contrast, low-skilled women face a labor market with limited opportunities for formal, full-time, and long-term employment, which significantly restricts their choices. As a result, traditional gender roles may become more pronounced among higher-skilled workers, where career aspirations and societal expectations intersect, and where individuals may have the flexibility to choose between a formal full-time or more flexible job arrangements.

To understand these dynamics better, we examine the distinct labor market patterns among individuals with varying levels of educational attainment.

## Results

Table 3 presents the estimation results for equation (1) on labor force participation. Column (1) displays the baseline model, which includes the controls described in the Data and empirical strategy section. Additional controls—such as a cubic polynomial of the husband's income and dummy variables indicating the presence of a child aged 4 years or younger and a child aged 5–12 years—are incorporated in Columns (2) and (3). The full set of regression coefficients for all specifications is reported in Tables A10–A14 in S1 Appendix.

In the base model, $\beta_1$ is estimated at 0.074. This suggests that a 10-percentage point (p.p) increase in the probability that a woman earns more than her partner is associated with around 0.74 p.p. increase in her likelihood of participating in the labor market. Contrary to the male breadwinner hypothesis, we find a positive relation between potential relative income and labor force participation. The estimates remain stable and statistically significant even after including the cubic polynomial of the partner's income and the child-related indicators. The fact that the estimate remains robust to the inclusion of additional controls suggests that the observed association is not driven by omitted variable bias.

As previously discussed, opting out of the labor market can be a costly decision, particularly in the developing world. Consequently, in many developing countries, we can reasonably suppose that labor market adjustments often occur through informal or flexible jobs rather than complete withdrawal from the workforce. Thus, engaging in less secure employment may be a strategy women adopt to remain in the labor force while adhering to social norms, particularly the expectation that a wife should not earn more than her husband. To explore this hypothesis, we examine the relation of a woman's probability of earning more than her husband and her likelihood of entering the informal sector. For this analysis, we restrict our sample to employed women.

Using the basic specification, the estimated coefficient is −0.1, as shown in Table 4(a), Columns (1)-(3). This finding implies that a 10-percentage-point increase in the probability of a woman earning more than her partner is associated with a 1 percentage point decrease in her likelihood of participating in the formal sector.

A similar pattern emerges when analyzing part-time, Table 4(b) shows that a 10-percentage-point increase in this probability corresponds to a 0.6 percentage point increase in the likelihood of part-time employment. Here, part-time employment is defined as working no more than 36 hours per week, in accordance with the legal framework. In the Appendix (Table A1 in S1 Appendix), we re-estimate these results using alternative thresholds of 20, 25, 30 and 39 hours per week, finding that the results remain qualitatively consistent.

**Table 3. Potential relative income and female labor force participation.**

|  | (1) | (2) | (3) |
|---|---|---|---|
|  | LFP | LFP | LFP |
| Prob. wife earns more | 0.074*** | 0.079*** | 0.079*** |
|  | (0.0055) | (0.0069) | (0.0069) |
| Cubic husband's income | No | Yes | Yes |
| Children indicator | No | No | Yes |
| Observations | 129009 | 129009 | 129009 |
| $R^2$ | 0.110 | 0.110 | 0.113 |

Source: Authors' own elaboration based on ENEMDU.

Robust standard errors in parenthesis are clustered at demographic group level.

+ p<0.10, * p<0.05, ** p<0.01, *** p<0.001.

**Table 4. Potential relative income and precarious jobs.**

| (a) Formal | (1) | (2) | (3) |
|---|---|---|---|
| Prob. wife earns more | −0.14*** | −0.100*** | −0.10*** |
| | (0.0070) | (0.0071) | (0.0070) |
| Cubic husband's income | No | Yes | Yes |
| Children indicator | No | No | Yes |
| Observations | 69680 | 69680 | 69680 |
| $R^2$ | 0.343 | 0.345 | 0.346 |
| **(b) Part-time** | **(1)** | **(2)** | **(3)** |
| Prob. wife earns more | 0.066*** | 0.060*** | 0.060*** |
| | (0.0067) | (0.0081) | (0.0081) |
| Cubic husband's income | No | Yes | Yes |
| Children indicator | No | No | Yes |
| Observations | 69680 | 69680 | 69680 |
| $R^2$ | 0.086 | 0.086 | 0.088 |
| **(c) Tenure/Long-term** | **(1)** | **(2)** | **(3)** |
| Prob. wife earns more | −0.095*** | −0.11*** | −0.11*** |
| | (0.0060) | (0.0072) | (0.0072) |
| Cubic husband's income | No | Yes | Yes |
| Children indicator | No | No | Yes |
| Observations | 69680 | 69680 | 69680 |
| $R^2$ | 0.209 | 0.209 | 0.210 |

Source: Authors' own elaboration based on ENEMDU.

Robust standard errors in parenthesis are clustered at demographic group level.

+ $p < 0.10$, * $p < 0.05$, ** $p < 0.01$, *** $p < 0.001$.

Finally, the likelihood of securing a tenured or long-term contract also decreases as the probability of a woman earning more than her partner rises. According to Table 4(c), a 10-percentage-point increase in this probability is associated with a 1 percentage point decrease in the likelihood of holding a tenured or long-term contract.

As discussed in the Data and empirical strategy section, evidence suggests a divergence in how gender norms seem to affect women, based on their skill levels. On one hand, professional identity for more educated women seem to conflict frequently with traditional gender roles, leading to reduced adherence to these norms. On the other hand, low-skilled women, constrained by limited opportunities for formal employment, appear to face restricted choices that reinforce traditional roles. Consequently, if gender norms extend across all educational levels in Ecuador, societal expectations are likely to become most pronounced among the highly educated segment. To explore these dynamics further, we analyze labor market patterns across different levels of educational attainment, due to measurement error in informal earnings and assortative matching, we keep our interpretation of the results as descriptive rather than causal.

The results on labor force participation (LFP) appear to be primarily driven by low-skilled workers. As shown in Table 5(a), the positive relationship between the probability of a woman earning more than her partner and labor market participation is significant only for women without a university education. While these findings are not fully consistent with the gender identity hypothesis, they suggest the need to examine how women's potential income is distributed across various employment types. By focusing on less-protected job types as a key mechanism for adapting to gender norms, we propose an alternative framework for understanding the role of gender norms in developing countries.

In contrast, Table 5(b) shows that the relationship between the probability of a woman earning more than her partner and formal sector employment is significant only for women with secondary or university education. A similar trend is

**Table 5. Potential relative income and precarious jobs by educational level.**

| | (1) | (2) | (3) |
| --- | --- | --- | --- |
| | **Primary** | **Secondary** | **University** |
| (a) Labor Force Participation | | | |
| Prob. wife earns more | 0.12*** | 0.074*** | −0.0060 |
| | (0.0097) | (0.013) | (0.013) |
| Observations | 72903 | 32545 | 23561 |
| $R^2$ | 0.117 | 0.046 | 0.033 |
| (b) Formal Sector | | | |
| Prob. wife earns more | −0.031* | −0.13*** | −0.061*** |
| | (0.013) | (0.017) | (0.011) |
| Observations | 36012 | 16246 | 17422 |
| $R^2$ | 0.077 | 0.092 | 0.042 |
| (c) Part-time Job | | | |
| Prob. wife earns more | 0.020 | 0.076*** | 0.063*** |
| | (0.014) | (0.017) | (0.013) |
| Observations | 36012 | 16246 | 17422 |
| $R^2$ | 0.037 | 0.044 | 0.025 |
| (d) Tenure/Long-term Job | | | |
| Prob. wife earns more | −0.12*** | −0.10*** | −0.12*** |
| | (0.0098) | (0.017) | (0.018) |
| Observations | 36012 | 16246 | 17422 |
| $R^2$ | 0.033 | 0.036 | 0.074 |

Source: Authors' own elaboration based on ENEMDU.

Robust standard errors in parenthesis are clustered at demographic group level.

+ $p < 0.10$, * $p < 0.05$, ** $p < 0.01$, *** $p < 0.001$.

observed in Table 5(c), where the possible effect of gender norms on part-time employment is driven primarily by high-skilled workers. Finally, tenure or long-term contract employment displays consistent results across all educational levels, indicating that this outcome is less correlated with a specific skill level.

These findings suggest that the relationship between the probability of a woman earning more than her partner and job precarity is particularly significant for women who are more likely to aspire to formal, full-time employment, a pathway that appears less accessible for low-skilled workers.

Finally, we present several robustness checks in S1 Appendix. The construction of the variable Prob. Wife Earns More, as proposed by Bertrand *et al.* [9], assumes that a woman's potential earnings in the labor market follow the distribution of observed earnings among married women within her demographic group. However, this measure may be influenced by gender norms, particularly in demographic groups where married women constitute the majority. If gender norms causally affect labor supply, they could bias the construction of this independent variable. To address this potential bias, we follow Galván [13] and conduct a robustness check using an alternative measure of the probability that the wife earns more. This alternative measure relies exclusively on the earnings distribution of non-married and non-cohabiting women, ensuring that potential earnings are not influenced by the gender norms that may prevail among married women. While non-married women do not constitute a random benchmark (since selection into marriage may correlate with unobserved characteristics affecting earnings potential) this alternative distribution provides a useful robustness check. The results,

presented in Tables A2 and A3 in S1 Appendix, confirm that both methods for estimating potential earnings yield consistent conclusions regarding the probability that the wife earns more.

Additionally, an alternative gender norm may also influence our findings: the belief that "women are responsible for child-rearing, while men are the primary providers." To explore this possibility, we assess in the appendix whether the results hold when the sample is divided into two groups—women with children and women without children. It is important to note that fertility decisions are endogenous; therefore, the child split is interpreted as a robustness check rather than as a causal test. If the child-rearing norm plays a dominant role, we would expect differing outcomes between these groups. However, as shown in Tables A4–A6 in S1 Appendix, the results remain consistent across both groups.

Ecuador experienced steady economic growth from 2007 to 2015, followed by a period of stagnation between 2016 and 2022. The average annual growth rate of GDP per capita (in constant prices) was 3% from 2007 to 2015, increasing from $4,929 to $6,241. However, this growth slowed to just 0.5% between 2016 and 2022, with GDP per capita rising only marginally from $6,106 to $6,411 [46]. To account for these differences, we conducted separate regressions for each period. Tables A7–A9 in S1 Appendix in the appendix demonstrate that the results are consistent across both periods

## Conclusions

The aim of this work was to explore the male breadwinner hypothesis in Ecuador, i.e., the social prescription that "a man should earn more than his wife", with relation to female LFP. Following the analysis carried out by Bertrand *et al.* [9], we investigate the female labor force participation when a woman can potentially have a higher income than her husband. Contrary to previous studies, we find a positive relationship between the probability of a woman earning more than her husband and her engagement in the labor market. However, we also find that the association is differentiated according to women's educational attainment, which is correlated with the income level of the family. Most of the positive relationship is due to the effect on low-skilled, low income female workers.

Women with university attendance who are likely to earn more than their partners do not increase their LFP, but the probability of them holding a tenured, full-time job decreases. Quite differently, women with secondary education (high school attendance) are more likely to increase both their LFP and their precariousness, shifting to part-time and informal positions. Finally, women with only primary education and low wages do increase their LFP but do not increase precariousness, which is already high: for these female workers the possibility to access a formal job is generally reduced.

These results contradict a part of the literature, but may shed light on how social norms and gender identity intertwine with income level and poverty. Consistent with the economic literature on individual choices under subsistence constraints, when a household faces limited resources, absolute priority is given to meeting basic needs. Only after surpassing the subsistence threshold do individuals consider trade-offs with other goods: this approach has been applied to climate and environmental issues, but it holds also for other intangible elements such as cultural services or adherence to social norms [47,48]. Our findings suggest that, in developing countries with fragmented labor markets, weak social protection and high poverty rates, like Ecuador, the need to secure a sufficient income for the family seems to act as a powerful driver of female employment. It must be considered that the Minimum Expenditure Basket for a household in Ecuador was USD 789,57 in January 2024 [49], compared to a minimum wage of USD 460: under such circumstances, women are forced to work, as just one salary for household in not enough to grant basic needs fulfilments.

Previous studies has already found that an increase in household incomes contributes negatively to female LFP in relatively poorer countries, arguing that an important share of women shape their LFP work out of economic necessity and choose their economic behavior accordingly [50,51]. In this light, adherence to social gender norms seems to be a "luxury", that poorer women (and couples) cannot afford. When the household is constrained by limited resources, weak socio-economic conditions and women face low access to formal and protected jobs, social norms' effect seem to weaken and the strategy to maximize income prevails, pushing women into the labor market.

These results have important policy implications. A deeper understanding of the drivers of female labor force participation is crucial for promoting a more comprehensive and informed policy discussion on gender and employment. Reducing gender disparities requires that policies take into account both cultural/social dimensions and the socio-economic conditions that characterize women and their environment: women may be trapped between fulfilling social expectations around their role within the household and the necessity to contribute to the family's basic needs. Policies targeting women should address gender discrimination in all its aspects and promote measures favoring formalization of female-dominated informal sectors (tax incentives for formal hiring, simplified business registration procedures for women-led microenterprises, and the implementation of portable benefits systems decoupled from standard employment contracts). Also, it is important to invest in policies aimed at alleviating women' child-caring and household burden, such as care infrastructure, expanding community-based childcare services and extending non-contributory parental benefits to informal workers. These interventions can be effective in reducing the care-related barriers to labor market participation and promote shared household responsibilities. However, to increase effectiveness, they must be complemented by actions that directly challenge the persistence of restrictive gender norms; some example may be public awareness campaigns, inclusive school curricula, and community-based dialogues involving both women and men, which can shift perceptions around caregiving, income distribution, and the value of women's work, both within and outside the household.

This study has also some limitation, which must be mentioned. Although the empirical strategy follows Bertrand et al. [9] and related work [12,13], and includes a rich set of individual, household, and job-related controls, marriage-market sorting may still confound the interpretation of relative income effects. In particular, women who anticipate weaker attachment to the labor market may sort into partnerships with lower-earning men, generating a correlation between the probability of earning more than the husband and observed labor supply outcomes, especially among women with primary and secondary education, that does not necessarily reflect normative responses.

Second, unobserved individual characteristics, such as ability, ambition, or occupation-specific preferences, may jointly influence potential earnings and employment outcomes. While the construction of potential income at the demographic-group level and the inclusion of detailed controls mitigate this concern, they cannot fully eliminate bias arising from unobserved heterogeneity.

Third, measurement error in earnings, especially in informal employment, may be more pronounced among low-skilled workers, which could affect comparisons across educational groups. Finally, in Ecuador's highly informal and segmented labor market, outcomes such as part-time or informal employment may reflect labor demand constraints or institutional features rather than purely voluntary adjustments driven by social norms. As a result, our findings should be interpreted as descriptive associations that are consistent with the role of gender norms operating through different labor market margins, rather than as causal estimates that cleanly separate supply- and demand-side mechanisms.

Future studies may add empirical evidence on how socially shaped gender identity impacts female engagement in the labor market, according to a country's level of development. Also, further inquiries may explore the interplay of social norms with other kinds of female discrimination; for instance, it would be interesting to investigate whether those low income women that take advantage of their possibility to earn more than their husbands are faced with other types of discrimination, such as higher domestic violence.

## Supporting information

**S1 Appendix. Supplementary estimations.**
(DOCX)

## Author contributions

**Conceptualization:** Jorge Yepez, Sara Caria.

**Data curation:** Jorge Yepez, Sara Caria.

**Formal analysis:** Jorge Yepez, Sara Caria.

**Investigation:** Jorge Yepez, Sara Caria.

**Methodology:** Jorge Yepez, Sara Caria.

**Software:** Jorge Yepez, Sara Caria.

**Writing – original draft:** Jorge Yepez, Sara Caria.

**Writing – review & editing:** Jorge Yepez, Sara Caria.

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
