## [Decision Letter · Decision Letter 0]

28 Jan 2026

PONE-D-25-65430Social norms vs socioeconomic vulnerability: gender identity and female labor force participation in EcuadorPLOS One

Dear Dr. Yepez,

Thank you for submitting your manuscript to PLOS ONE. After careful consideration, we feel that it has merit but does not fully meet PLOS ONE’s publication criteria as it currently stands. Therefore, we invite you to submit a revised version of the manuscript that addresses the points raised during the review process.

We look forward to receiving your revised manuscript.

Kind regards,

José Alberto Molina

Academic Editor

PLOS One

Journal Requirements:

Reviewers' comments:

Reviewer's Responses to Questions

**Comments to the Author**

1. Is the manuscript technically sound, and do the data support the conclusions?

Reviewer #1: Partly

Reviewer #2: Yes

2. Has the statistical analysis been performed appropriately and rigorously? 

Reviewer #1: Yes

Reviewer #2: Yes

3. Have the authors made all data underlying the findings in their manuscript fully available?

Reviewer #1: Yes

Reviewer #2: Yes

4. Is the manuscript presented in an intelligible fashion and written in standard English?

Reviewer #1: Yes

Reviewer #2: Yes

5. Review Comments to the Author

Reviewer #1: This paper examines how gender norms relate to married women’s labor market outcomes in Ecuador using an adaptation of the probability-based measure from Bertrand et al. (2015). The authors construct a counterfactual indicator of whether a wife would outearn her husband, conditional on demographic characteristics, and examine its association with labor force participation, formality, working hours, and job stability. The paper also explores heterogeneity by educational attainment and provides several robustness checks.

The empirical strategy is transparent, replicable, and grounded in previously published methodology. The results are well organized, coherently discussed, and contribute descriptive evidence to our understanding of gender norms in a highly informal labor market setting. The heterogeneity analysis adds depth and policy relevance.

Major Comments

Interpretation. The baseline positive association with labor force participation (LFP) is interpreted as evidence against breadwinner norms. However, the constructed regressor captures both potential earning capacity and exposure to norms. Standard labor supply responses (higher potential earnings → higher LFP) could generate similar patterns. The paper should avoid causal or mechanism language (e.g., “impact”, “effect”) and frame results as associations or patterns consistent with certain hypotheses.

Heterogeneity. The education-split results are informative: LFP associations are driven by low-skilled women, while intensive-margin outcomes (formality, full-time work, tenure) are more pronounced for high-skilled women. This helps interpret the baseline coefficient. However, measurement error in informal earnings and assortative matching by education may confound these comparisons and should be acknowledged.

Identification and Limitations. While the empirical strategy follows recent literature, several identification issues remain. Potential confounding arises from:

a) Marriage market sorting: Women who anticipate lower labor force attachment may sort into marriages with lower-earning men, generating a mechanical correlation between the breadwinner probability and labor supply choices that does not reflect normative behavior.

b) Unobserved heterogeneity: Ability, ambition, and occupation-specific preferences affect both potential earnings and employment outcomes but are not directly observed. Although the controls reduce this concern, they do not eliminate it.

c) Labor demand constraints: In Ecuador’s highly informal labor market, observed outcomes such as part-time or informal work may reflect labor demand or institutional constraints rather than normative adaptation. Without separating supply from demand channels, causal attribution remains unclear.

These issues do not invalidate the descriptive analysis but should be made explicit in a short limitations section. The current draft hints at these issues but does not present them directly and transparently.

Reproducibility. To enhance transparency, the authors should clarify (i) whether earnings distributions are computed by year, (ii) how informal earnings are measured, and (iii) how missing data are treated. Providing replication code would further improve reproducibility.

Robustness. The paper includes thoughtful robustness exercises (alternative earnings distributions, child splits, macro periods). The authors should note that non-married women are not a random benchmark, and anticipation effects may limit child-split interpretation.

Minor Comments

Briefly justify the clustering level for standard errors.

Standardize terminology (e.g., “formal sector”, “full-time”, “tenure”).

The literature review section could be more concise.

Reviewer #2: This paper investigates gender norms on Ecuador’s labor market, specifically examining how the "male breadwinner" social norm correlates with labor force participation, formal versus informal employment, full-time versus part-time status, and job tenure.

This paper is methodologically sound and effectively builds upon established literature, adding to its credibility/robustness. The robustness checks are sound and further contribute to demonstrating the replicability of these findings across multiple specifications.

In the following lines, I recommend a series of minor revisions aimed at improving the transparency and clarity of certain sections.

1. In line with the principles of open science, I recommend the authors share their analysis code via a public repository (e.g., OSF). This practice enhances transparency and facilitates a more efficient, cumulative scientific process.

2. I recommend reporting the full set of regression coefficients for all variables included in the models. Providing these estimates—rather than only the primary variables of interest—allows readers to more accurately assess the model specification and identify potential issues.

3. The manuscript defines full-time employment as requiring at least 40 hours (page 9), yet references a 36-hour threshold for part-time status (page 10). The authors should clarify how workers falling between 36 and 40 hours are classified. If they are not classified as either full-time or part-time employees, the authors should clarify how their findings apply to them.

4. The first paragraph of page 9 provides an ambiguous explanation of the economic rationale behind this study. I recommend the authors revise this section to more precisely present the theoretical framework, ensuring that it aligns with the empirical design implemented in the analyses.

5. Regarding the linear probability model in Equation 2, the authors should clarify how the controls for "wife’s potential income at each 5th percentile" are constructed. It is unclear whether these variables represent the mean income, the maximum income, or another measure of income within the 5-percentile band.

6. The manuscript contains typos. For instance, at the end of page 15, the authors state "the estimate is reduced to -0.6," (should be -0.06). Similarly, in page 20 the authors write “a woman earning 2more than her husband”. I encourage the authors to carefully review the entire text.

6. PLOS authors have the option to publish the peer review history of their article (what does this mean?). If published, this will include your full peer review and any attached files.

Reviewer #1: No

Reviewer #2: No

---

## [Author Response · Author response to Decision Letter 1]

27 Mar 2026

Response to reviewers

Paper: Social norms vs socioeconomic vulnerability: gender identity and female labor force participation in Ecuador

We would like to thank the Editor and both Reviewers for their careful reading of our manuscript and for their valuable and constructive comments. We found the suggestions helpful in improving the clarity, positioning, and robustness of the paper. In the revised version, we have tried to carefully address all points raised by the Reviewers. We also conducted a general revision, correcting some minor typos in the manuscript. Below, we provide a point-by-point response to the issues raised. For ease of reference, we first reproduce each comment and then provide our response.

Response to Reviewer #1

This paper examines how gender norms relate to married women’s labor market outcomes in Ecuador using an adaptation of the probability-based measure from Bertrand et al. (2015). The authors construct a counterfactual indicator of whether a wife would outearn her husband, conditional on demographic characteristics, and examine its association with labor force participation, formality, working hours, and job stability. The paper also explores heterogeneity by educational attainment and provides several robustness checks.

Answer: The empirical strategy is transparent, replicable, and grounded in previously published methodology. The results are well organized, coherently discussed, and contribute descriptive evidence to our understanding of gender norms in a highly informal labor market setting. The heterogeneity analysis adds depth and policy relevance.

Thank you very much. We are grateful for this positive general comment on our work and also for the thoughtful and useful comments detailed below.

Major Comments

Interpretation. The baseline positive association with labor force participation (LFP) is interpreted as evidence against breadwinner norms. However, the constructed regressor captures both potential earning capacity and exposure to norms. Standard labor supply responses (higher potential earnings → higher LFP) could generate similar patterns. The paper should avoid causal or mechanism language (e.g., “impact”, “effect”) and frame results as associations or patterns consistent with certain hypotheses.

Answer: We thank the reviewer for this correct comment. We have revised the text according to the suggestion and reframed our interpretation in terms of results consistent with the male breadwinner hypothesis. We have also better clarified which assumptions we build on.

Heterogeneity. The education-split results are informative: LFP associations are driven by low-skilled women, while intensive-margin outcomes (formality, full-time work, tenure) are more pronounced for high-skilled women. This helps interpret the baseline coefficient. However, measurement error in informal earnings and assortative matching by education may confound these comparisons and should be acknowledged.

Answer: We thank the reviewer for this insightful comment. We agree that heterogeneity by education is an important dimension of our results. We also acknowledge that measurement error in informal earnings and assortative matching by education may confound these comparisons. We now note these limitations in the discussion to clarify the interpretation of the baseline coefficient.

“due to measurement error in informal earnings and assortative matching, we keep our interpretation of the results as descriptive rather than causal.’

Identification and Limitations. While the empirical strategy follows recent literature, several identification issues remain. Potential confounding arises from:

a) Marriage market sorting: Women who anticipate lower labor force attachment may sort into marriages with lower-earning men, generating a mechanical correlation between the breadwinner probability and labor supply choices that does not reflect normative behavior.

b) Unobserved heterogeneity: Ability, ambition, and occupation-specific preferences affect both potential earnings and employment outcomes but are not directly observed. Although the controls reduce this concern, they do not eliminate it.

c) Labor demand constraints: In Ecuador’s highly informal labor market, observed outcomes such as part-time or informal work may reflect labor demand or institutional constraints rather than normative adaptation. Without separating supply from demand channels, causal attribution remains unclear.

These issues do not invalidate the descriptive analysis but should be made explicit in a short limitations section. The current draft hints at these issues but does not present them directly and transparently.

Answer: We fully acknowledge that comparisons across educational groups may be affected by several factors that cannot be fully addressed in the paper. In response to this comment, we have added the following paragraph in the Conclusions section, where we explicitly discuss marriage market sorting, unobserved heterogeneity, measurement error in informal earnings, and labor demand constraints. We believe this improves transparency and clarifies the scope of our empirical claims.

“This study has also some limitation, which must be mentioned. Although the empirical strategy follows Bertrand et al. (2015) and related work (e.g., Codazzi et al., 2018; Galván, 2022), and includes a rich set of individual, household, and job-related controls, marriage-market sorting may still confound the interpretation of relative income effects. In particular, women who anticipate weaker attachment to the labor market may sort into partnerships with lower-earning men, generating a correlation between the probability of earning more than the husband and observed labor supply outcomes, especially among women with primary and secondary education, that does not necessarily reflect normative responses.

Second, unobserved individual characteristics, such as ability, ambition, or occupation-specific preferences, may jointly influence potential earnings and employment outcomes. While the construction of potential income at the demographic-group level and the inclusion of detailed controls mitigate this concern, they cannot fully eliminate bias arising from unobserved heterogeneity.

Third, measurement error in earnings, especially in informal employment, may be more pronounced among low-skilled workers, which could affect comparisons across educational groups. Finally, in Ecuador’s highly informal and segmented labor market, outcomes such as part-time or informal employment may reflect labor demand constraints or institutional features rather than purely voluntary adjustments driven by social norms. As a result, our findings should be interpreted as descriptive associations that are consistent with the role of gender norms operating through different labor market margins, rather than as causal estimates that cleanly separate supply- and demand-side mechanisms.”

Reproducibility. To enhance transparency, the authors should clarify (i) whether earnings distributions are computed by year, (ii) how informal earnings are measured, and (iii) how missing data are treated. Providing replication code would further improve reproducibility.

Answer: We thank the reviewer for this helpful comment and for highlighting the importance of transparency and reproducibility.

(i) Earnings distributions by year: Figure 1 is constructed using pooled data. To clarify this point and assess robustness, we now mention in the manuscript that year-specific relative income distributions and corresponding McCrary tests are reported in the Appendix, which display the same qualitative pattern, albeit with reduced statistical significance due to smaller sample sizes.

(ii) Measurement of informal earnings: The analysis uses the ENEMDU microdata, publicly available from the Ecuadorian National Institute of Statistics and Census (INEC). Labor income is the monthly labor earnings variable provided in the data, including main and secondary occupations (wage earners, self-employed, employers, domestic workers) and excluding capital income, transfers, and social benefits. We clarify this in the new version.

(iii) Treatment of missing values: Missing values occur only for the husband’s labor income, since female labor income is not directly used as a regressor. Observations with missing husband income are excluded from the estimation sample. We have clarified this point in the revised version of the paper

In line with open science principles, we have uploaded all replication code to our public OSF repository: https://doi.org/10.17605/OSF.IO/K8WRD The link is now provided in the revised manuscript to enhance transparency and reproducibility.

Robustness. The paper includes thoughtful robustness exercises (alternative earnings distributions, child splits, macro periods). The authors should note that non-married women are not a random benchmark, and anticipation effects may limit child-split interpretation.

Answer: We thank the reviewer for this important remark. We agree that non-married women do not constitute a random benchmark group, as selection into marriage may be correlated with unobserved characteristics that also affect earnings potential. We acknowledge this in the revised version of the paper.

“While non-married women do not constitute a random benchmark—since selection into marriage may correlate with unobserved characteristics affecting earnings potential—this alternative distribution provides a useful robustness check.”

We agree with the reviewer that fertility decisions are not exogenous, as women may anticipate future childbearing when making labor market choices. We have clarified this point in the revised manuscript to ensure that the child-split results are interpreted as descriptive robustness checks rather than as causal evidence.

“It is important to note that fertility decisions are endogenous; therefore, the child split is interpreted as a robustness check rather than as a causal test.”

Minor Comments

Briefly justify the clustering level for standard errors.

Answer: We thank the reviewer for the comment. Following Galván (2022) and related literature, we cluster standard errors at the wife’s demographic group level to account for potential correlation of outcomes within groups defined by age, education, region, and ethnicity. This clustering captures unobserved heterogeneity that could affect labor market outcomes within these demographic groups. We included this on the paper.

“Standard errors are clustered at the wife’s demographic group level to account for potential within-group correlation in labor market outcomes, following Galván (2022).”

Standardize terminology (e.g., “formal sector”, “full-time”, “tenure”).

Answer: We thank the reviewer for this useful comment. The terminology we adopted is the official one used by the INEC, with whom non-Ecuadorian readers may not be familiar with. We have detailed what each term refers to, according to the legislation.

The literature review section could be more concise.

Answer: Thank you for this comment. We have reduced the references and the literature review excluding those debates that only marginally touch upon our topic. We have maintained only those streams that contribute directly to the framing of our study, that’s to say the ones referring to how social norms may influence women economic behaviuor.

Response to Reviewer #2:

This paper investigates gender norms on Ecuador’s labor market, specifically examining how the "male breadwinner" social norm correlates with labor force participation, formal versus informal employment, full-time versus part-time status, and job tenure.

This paper is methodologically sound and effectively builds upon established literature, adding to its credibility/robustness. The robustness checks are sound and further contribute to demonstrating the replicability of these findings across multiple specifications.

Answer: Thank you very much. We are grateful for this positive general comment on our work and also for the thoughtful and useful comments detailed below.

In the following lines, I recommend a series of minor revisions aimed at improving the transparency and clarity of certain sections.

1. In line with the principles of open science, I recommend the authors share their analysis code via a public repository (e.g., OSF). This practice enhances transparency and facilitates a more efficient, cumulative scientific process.

Answer: We thank the reviewer for this helpful suggestion. In line with open science principles, we have uploaded all replication code to our public OSF repository:https://doi.org/10.17605/OSF.IO/K8WRD . The link is now provided in the revised manuscript to enhance transparency and reproducibility.

2. I recommend reporting the full set of regression coefficients for all variables included in the models. Providing these estimates—rather than only the primary variables of interest—allows readers to more accurately assess the model specification and identify potential issues.

Answer: We thank the reviewer for this helpful suggestion. In response, we now report the full set of regression coefficients for all model specifications in Appendix A4. Presenting the complete results allows readers to fully assess the specification, evaluate the role of control variables, and examine the robustness of the estimates beyond the primary coefficients of interest. We mentioned this in the new version of the manuscript.

3. The manuscript defines full-time employment as requiring at least 40 hours (page 9), yet references a 36-hour threshold for part-time status (page 10). The authors should clarify how workers falling between 36 and 40 hours are classified. If they are not classified as either full-time or part-time employees, the authors should clarify how their findings apply to them.

Answer: We thank the reviewer for this comment. The reviewer correctly notes that our initial definition of “full employment,” which is specific to Ecuador and includes both earnings above the minimum wage and working at least 40 hours per week, could cause confusion. To avoid this local-specific classification, in the revised version we removed the full-employment classification and retained only the part-time classification, defined as working ≤36 hours per week according to Ecuadorian labor law. We belive that this definition is more universal and easier for readers to interpret. As a robustness check, we also re-estimated the results using alternative thresholds for part-time employment (≤20, ≤25, ≤30, and ≤39 hours at the Appendix), finding consistent results. We believe these changes improve clarity and readability, and we thank the reviewer for this helpful suggestion.

4. The first paragraph of page 9 provides an ambiguous explanation of the economic rationale behind this study. I recommend the authors revise this section to more precisely present the theoretical framework, ensuring that it aligns with the empirical design implemented in the analyses.

Answer: We thank the reviewer for this suggestion. We have expanded the explication on the economic rational we start from. We have integrated a simple model to represent a female’s utility function, drawn by Bertrand et al (2015), and as also used in Codazzi et al. (2018). This model illustrates the theoretical hypothesis that guides our empirical design, making explicit the link between social norms, relative earnings, and women’s labor supply decisions.

5. Regarding the linear probability model in Equation 2, the authors should clarify how the controls for "wife’s potential income at each 5th percentile" are constructed. It is unclear whether these variables represent the mean income, the maximum income, or another measure of income within the 5-percentile band.

Answer: We thank the reviewer for this comment. The controls for the wife’s potential income at each 5th percentile represent the actual percentile value, as constructed in the data for each demographic group. This approach follows the methodology used by Galván (2022) and Codazzi (2018). We clarify this in the new version.

“the wife’s potential income at each 5th percentile (corresponding to the actual percentile within her dem

---

## [Decision Letter · Decision Letter 1]

30 Apr 2026

Social norms vs socioeconomic vulnerability: gender identity and female labor force participation in Ecuador

PONE-D-25-65430R1

Dear Dr. Yepez,

We’re pleased to inform you that your manuscript has been judged scientifically suitable for publication and will be formally accepted for publication once it meets all outstanding technical requirements.

Kind regards,

José Alberto Molina

Academic Editor

PLOS One

Additional Editor Comments (optional):

Reviewers' comments:

Reviewer's Responses to Questions

**Comments to the Author**

1. If the authors have adequately addressed your comments raised in a previous round of review and you feel that this manuscript is now acceptable for publication, you may indicate that here to bypass the “Comments to the Author” section, enter your conflict of interest statement in the “Confidential to Editor” section, and submit your "Accept" recommendation.

Reviewer #1: All comments have been addressed

Reviewer #2: All comments have been addressed

2. Is the manuscript technically sound, and do the data support the conclusions?

Reviewer #1: Yes

Reviewer #2: (No Response)

3. Has the statistical analysis been performed appropriately and rigorously? 

Reviewer #1: Yes

Reviewer #2: (No Response)

4. Have the authors made all data underlying the findings in their manuscript fully available?

Reviewer #1: Yes

Reviewer #2: (No Response)

5. Is the manuscript presented in an intelligible fashion and written in standard English?

Reviewer #1: Yes

Reviewer #2: (No Response)

6. Review Comments to the Author

Reviewer #1: (No Response)

Reviewer #2: (No Response)

7. PLOS authors have the option to publish the peer review history of their article (what does this mean?). If published, this will include your full peer review and any attached files.

Reviewer #1: No

Reviewer #2: No

---

## [Editor Report · Acceptance letter]

PONE-D-25-65430R1

PLOS One

Dear Dr. Yepez,

I'm pleased to inform you that your manuscript has been deemed suitable for publication in PLOS One. Congratulations! Your manuscript is now being handed over to our production team.

Kind regards,

on behalf of

Professor José Alberto Molina

Academic Editor

PLOS One